# Beneficial Effect of Cuban Policosanol on Blood Pressure and Serum Lipoproteins Accompanied with Lowered Glycated Hemoglobin and Enhanced High-Density Lipoprotein Functionalities in a Randomized, Placebo-Controlled, and Double-Blinded Trial with Healthy Japanese

**DOI:** 10.3390/ijms24065185

**Published:** 2023-03-08

**Authors:** Kyung-Hyun Cho, Hyo-Seon Nam, Seung-Hee Baek, Dae-Jin Kang, Hyejee Na, Tomohiro Komatsu, Yoshinari Uehara

**Affiliations:** 1Raydel Research Institute, Medical Innovation Complex, Daegu 41061, Republic of Korea; 2LipoLab, Yeungnam University, Gyeongsan 38541, Republic of Korea; 3Center for Preventive, Anti-Aging and Regenerative Medicine, Fukuoka University Hospital, 8-19-1 Nanakuma, Johnan-ku, Fukuoka 814-0180, Japan; 4Faculty of Sports and Health Science, Fukuoka University, 8-19-1 Nanakuma, Johnan-ku, Fukuoka 814-0180, Japan

**Keywords:** policosanol, low-density lipoproteins (LDL), high-density lipoproteins (HDL), glycation, oxidation, HDL functionality, apolipoprotein A-I, human trial, zebrafish

## Abstract

This study evaluated the efficacy and safety of 20 mg of Cuban policosanol in blood pressure (BP) and lipid/lipoprotein parameters of healthy Japanese subjects via a placebo-controlled, randomized, and double-blinded human trial. After 12 weeks of consumption, the policosanol group showed significantly lower BP, glycated hemoglobin (HbA_1c_), and blood urea nitrogen (BUN) levels. The policosanol group also showed lower aspartate aminotransferase (AST), alanine aminotransferase (ALT), and γ-glutamyl transferase (γ-GTP) levels at week 12 than those at week 0: A decrease of up to 9% (*p* < 0.05), 17% (*p* < 0.05), and 15% (*p* < 0.05) was observed, respectively. The policosanol group showed significantly higher HDL-C level and HDL-C/TC (%), approximately 9.5% (*p* < 0.001) and 7.2% (*p* = 0.003), respectively, than the placebo group and a difference in the point of time and group interaction (*p* < 0.001). In lipoprotein analysis, the policosanol group showed a decrease in oxidation and glycation extent in VLDL and LDL with an improvement of particle shape and morphology after 12 weeks. HDL from the policosanol group showed in vitro stronger antioxidant and in vivo anti-inflammatory abilities. In conclusion, 12 weeks of Cuban policosanolconsumption in Japanese subjects showed significant improvement in blood pressure, lipid profiles, hepatic functions, and HbA_1c_ with enhancement of HDL functionalities.

## 1. Introduction

Policosanol is a mixture of aliphatic alcohols ranging from 24–34 carbon atoms [1,2], such as octacosanol, triacontanol, dotriacontanol, hexacosanol, and tetratriacontanol, which are purified from sugar cane (*Saccharum officinarum* L.) wax [1,2,3] or various plants, such as oats [4], barley [5], insects [6,7], and bees wax [8]. Many policosanols from different sources have been claimed to treat blood dyslipidemia [9,10], diabetes [11], hypertension [12,13], and dementia [14,15] by raising the HDL-C level and lowering the LDL-C level. However, with the exception of Cuban policosanol [12,16,17], there is insufficient information on policosanol regarding its compositions and physiological effects on the lipoprotein metabolism, particularly in HDL functionality. In addition to the increase in HDL-C quantity, the improvement of HDL functionality should be accompanied to maximize the efficacy of policosanol. The HDL functionality in blood, such as antioxidant and anti-inflammatory properties, should be improved by policosanol consumption [18,19] since dysfunctional HDL is more atherogenic and exacerbates the proinflammatory cascade [20].

Many studies examining the molecular mechanisms for the efficacy of Cuban policosanol reported that the encapsulation of policosanol into HDL enhances the HDL functions and has anti-senescence and tissue regeneration effects by improving anti-glycation, anti-apoptosis, and cholesteryl transfer protein (CETP) inhibition [16]. CETP is an HDL-associated protein that contributes to HDL remodeling: Lowers the HDL-C and raises the triglyceride contents in HDL, which is associated with the production of dysfunctional HDL [21]. Cuban policosanol supplementation for 9 weeks in zebrafish had serum lipid-lowering and HDL-C-elevating effects via CETP inhibition [22]. Policosanol supplementation for 8 weeks also ameliorated the fatty liver changes with less production of reactive oxygen species (ROS) in zebrafish and rats [22,23]. Cuban policosanol supplementation in Korean participants increased the serum HDL-C and enhanced the HDL functionality to inhibit the oxidation and glycation of LDL and HDL [17,18]. The consumption of policosanol for 8 weeks by healthy female subjects with pre-hypertension resulted in lower blood pressure (BP) and CETP ability by elevating the HDL/apoA-I contents and enhancing the HDL functionalities, including cholesterol efflux and insulin secretion [18]. Eight weeks of policosanol supplementation in spontaneously hypertensive rats (SHR) resulted in remarkable dose-dependent decreases in BP [23]. In addition to increasing the HDL-C level after 12 weeks of consumption [12], long-term (24 weeks) policosanol consumption lowered the BP while enhancing the advantageous functions of HDL, including its antioxidant, anti-glycation, and anti-atherosclerotic activities [18].

Although many studies reported the efficacy of policosanol in humans, particularly in a Korean population, there has been no study on the effects of Cuban policosanol on the lipid parameters of a Japanese population. The Japanese population shows the unique features of a higher HDL-C level than western populations, with a higher portion of those with a genetic deficiency of CETP [24]. In a similar period between 2000 and 2002, Japanese populations showed higher HDL-C levels (~55 mg/dL) than American (United States) populations (46 mg/dL) [25,26], while Korean populations showed the lowest level of HDL-C (~43 mg/dL) [27,28]. These differences in serum HDL-C levels depend on the country. Moreover, ethnicities might influence the efficacy of policosanol in improving the lipid, lipoprotein profile, and HDL functionality.

The current study examined the effect of policosanol consumption on the blood lipid parameters of a Japanese population; healthy Japanese subjects who had a normal blood pressure (BP) and normolipidemic (120 mg/dL < LDL < 160 mg/dL and >40 mg/dL of HDL) were recruited. The participants were randomized to consume 20 mg of policosanol or a placebo to compare the changes in BP, blood parameters, and lipid and lipoprotein parameters with a randomized and double-blinded test. After 12 weeks of consumption of policosanol or a placebo, blood was analyzed to assess the putative efficacy to improve the metabolic parameters of the heart, kidney, liver, or hidden toxicity. From the participants, HDL and LDL were purified individually, and the HDL functionality and LDL qualities, such as oxidized and glycated extent with antioxidant abilities, were analyzed.

The antioxidant and anti-inflammatory properties of HDL were compared using zebrafish embryos by testing their developmental speed and survivability after injecting HDL in the presence of *N*-ε-carboxymethyllysine (CML), which is a proinflammatory and neurotoxin [29]. An elevated serum CML level is also associated with the exacerbation of atherosclerosis via lipoprotein modifications and increased susceptibility to low-density lipoproteins (LDL) oxidation [30]. Higher CML serum levels were associated with high-sensitivity C-reactive protein (CRP) via an increase in toll-like receptor 4 (TLR-4) expression in monocytes [31,32].

Zebrafish (*Danio rerio*) is a widely used vertebrate model to test the putative anti-inflammatory effects of drug candidates since zebrafish embryos have well-developed innate and acquired immune systems similar to the mammalian immune system [33]. An additional advantage of working with zebrafish embryos is that they develop externally and are optically transparent during development. With these characteristics, zebrafish and their embryos are a valuable and popular animal model for various studies, including inflammation [34].

The improvements of the quantity and quality of lipoproteins in blood are very important for evaluating the efficacy of policosanol or any nutraceutical to treat dyslipidemia and hypertension. In the current study, changes in the BP, lipid/lipoprotein parameters, and HDL functionalities were assessed after consuming Cuban policosanol for 12 weeks, using a randomized, double-blinded, placebo-controlled study.

## 2. Results

### 2.1. Anthropometric and Blood Profiles

As shown in Table 1, the two groups showed no difference in body mass index (BMI) and heart rate between weeks 0 and 12. The policosanol group (n = 30) showed a 7.1% (*p* < 0.001) and 4.0 % (*p* = 0.034) decrease in systolic blood pressure (SBP) and diastolic blood pressure (DBP) at week 12, compared with week 0. On the other hand, the placebo group showed no changes in the SBP and DBP between weeks 0 and 12, despite all participants showing a normotensive range at week 0. Interestingly, the glycated hemoglobin (HbA_1c_) level was 4% lower at week 12 than week 0 in the policosanol group (*p* = 0.009), while the placebo group showed a similar HbA_1c_ level between weeks 0 and 12. In a group comparison during 12 weeks, the glycated hemoglobin level was 2% lower in the policosanol group than in the placebo group (*p* = 0.024). These results suggest that policosanol consumption for 12 weeks can help in lowering the BP and HbA_1c_ simultaneously without changing the BMI and heart rate. The other blood parameters (total protein, albumin, albumin/globulin (A/G) ratio, uric acid, and glucose) were relatively unaffected between weeks 0 and 12 in the policosanol and placebo groups. These results suggest that policosanol consumption did not affect the nutrient metabolism in protein, purine, and carbohydrate homeostasis. 

### 2.2. Liver, Kidney, and Inflammatory Parameters

At week 12, the policosanol group showed 8.7% lower (*p* = 0.022) aspartate aminotransferase (AST) levels and 17.0% lower (*p* = 0.013) alanine aminotransferase (ALT) levels than those of week 0. In contrast, the placebo group did not show notable changes in the enzyme levels (Table 1). Interestingly, at week 12, the policosanol group showed a 15.4% (*p* = 0.016) and 6.0% (*p* = 0.052) larger decrease in γ-GTP and blood urea nitrogen (BUN) than at week 0, while the placebo group did not show a notable change. The total bilirubin level of the placebo group at week 12 was 11.8% higher than at week 0 (*p* = 0.030), while the policosanol group showed no change in the total bilirubin between weeks 0 and 12. These results suggest that policosanol consumption helps in the protection from liver damage and does not have liver toxicity.

At week 12, the placebo group had even higher γ-GTP and BUN levels than at week 0, even though no significance was detected. The placebo group showed a 16% higher BUN level than the policosanol group at week 12 (*p* = 0.001), even though they showed a similar level at week 0. In the policosanol and placebo groups, the creatinine, high sensitivity C-reactive protein (hsCRP), and lactate dehydrogenase (LDH) levels were similar at weeks 0 and 12, indicating that policosanol consumption did not affect acute inflammation and endocrinological damage in the liver and kidney.

As listed in Table 1, the policosanol group at week 12 showed a 10% increase in the serum apoA-I level compared with week 0, from 165 ± 2 mg/dL to 182 ± 3 mg/dL according to a paired *t*-test (*p* = 0.045), whereas the placebo group showed no change from 164 ± 5 mg/dL (week 0) to 160 ± 2 mg/dL (week 12). Analysis of covariance (ANCOVA) during 12 weeks revealed that the policosanol group showed 14% higher apoA-I levels than the placebo group (*p* = 0.028). On the other hand, the apo-B level in both groups was unchanged during the 12 weeks, around 98–101 mg/dL.

### 2.3. Lipid and Lipoprotein Profiles

After excluding the participants showing low compliance, who consumed a significantly more fat diet, heavy drinking, and smoking, during the 12-week consumption, the policosanol group (n = 15) showed 6.3% higher HDL-C (*p* = 0.006) at week 12 than week 0 (Table 2). In contrast, the placebo group (n = 17) showed a 6.6% decrease in the HDL-C level at week 12 from the baseline, week 0. At week 12, the policosanol group showed a 9.5% higher HDL-C level (*p* < 0.001) than the placebo group (n = 17). Repeated measures ANOVA of HDL-C showed that the policosanol group (n = 15) showed a significant difference from the placebo group (n = 17) in the point of time and group interaction (*p* < 0.001) during the 12 weeks. Interestingly, the policosanol group showed an 11% lower LDL-C level at week 8 (*p* = 0.013) than the placebo group. The LDL-C/HDL-C ratio was significantly lower in the policosanol group at weeks 8 (*p* = 0.018) and 12 than in the placebo group.

On the other hand, the TC and TG levels in the policosanol group were relatively unaffected at weeks 0, 4, 8, and 12. Repeated measures ANOVA of TC, TG, LDL-C, and RC levels during the 12 weeks showed no difference between the policosanol group (n = 15) and the placebo group (n = 17) in terms of time and group interaction. On the other hand, at week 12, the policosanol group showed a 3% lower TG/HDL-C ratio than the placebo group (*p* = 0.018) according to an analysis of covariance (ANCOVA) from baseline. The HDL-C/TC (%) ratio was increased in the policosanol group from 29.2 ± 1.1% to 30.9 ± 1.4% (*p* = 0.003) between weeks 0 and 12, while the placebo group showed a slight decrease from 29.7 ± 0.8% to 28.8 ± 1.0%. At week 12, the policosanol group showed 7.2% higher HDL-C/TC (%) than the placebo group (*p* = 0.003). Repeated measures ANOVA of HDL-C/TC (%) revealed a significant difference in the point of time and group interaction between the policosanol group (n = 15) and the placebo group (n = 17) (*p* < 0.033) during the 12 weeks, as shown in Table 2. These results suggest that HDL-C (mg/dL) and HDL-C/TC (%) were increased significantly by policosanol consumption at different times and group interactions according to repeated measures ANOVA.

### 2.4. VLDL Particle Observation and Composition Analysis

As shown in Figure 1, transmitted electron microscopy (TEM) showed that the particle number of VLDL in the policosanol group decreased at week 12 compared with week 0, while the placebo group showed a larger increase in particle number (Figure 1A). After 12 weeks of consumption, the VLDL particle size decreased 27% more in the policosanol group (*p* = 0.013) than at week 0, while the placebo group showed a 10% increase in particle size compared with week 0 (Figure 1B and Table 3).

As shown in Table 3, after 12 weeks of consumption, the policosanol group showed a 7% and 63% decrease in the extent of glycation (fluorescence intensity, FI) and oxidation (malondialdehyde, MDA) in VLDL, respectively, while the placebo group showed similar levels during the same period. The particle diameter of VLDL was significantly lower in the policosanol group at week 12 (~14% smaller) than at week 0, while the placebo group showed a 6% increase in diameter. In the policosanol group, the TC and TG contents increased by 45% and decreased by 15%, respectively, during the 12 weeks of consumption, whereas the placebo group showed around a 7% decrease and a 2% increase in the TC and TG content, respectively, in VLDL during the same period. These results suggest that policosanol consumption induced anti-atherogenic changes in the VLDL properties to exhibit lower glycation, oxidation, and TG content with a smaller particle size and number.

As shown in Figure 2A, native electrophoresis (0.5% agarose) under the nondenatured state of VLDL showed that native VLDL (lane 1), which was purified from young and healthy volunteers, showed more distinct band intensity than the oxVLDL band (lane 2), which was oxidized by a cupric ion treatment (final 10 μM). The oxVLDL band almost disappeared with the fastest electromobility due to the degradation of the apo-B band and the increase in the negative charges of VLDL (lane 2, Figure 2A). VLDL under a nondenatured state revealed that the policosanol group at week 12 (lane 4) had more distinct band intensity and slower electromobility than at week 0 (lane 3), which showed a larger smear and weaker band intensity. On the other hand, the placebo group showed a similar band intensity and electromobility between weeks 0 (lane 5) and 12 (lane 6), with almost no change in the TC and TG content in VLDL. The oxVLDL (lane 2) and VLDL at week 0 (lane 3) showed faster smear band intensities (Figure 2A) since the larger TG and MDA content in VLDL caused a larger smear band intensity with faster electromobility.

As shown in Figure 2B, quantification of oxidized species in VLDL showed that oxVLDL had the highest MDA level of 26 ± 3 μM MDA, while the native VLDL showed 7 ± 3 μM MDA. At week 0, both groups showed a similar level of MDA in VLDL around 12–15 μM MDA, as shown in Figure 2B. On the other hand, at week 12, the policosanol group (5.5 ± 1.5 μM MDA) showed 63% lower MDA levels than at week 0 (*p* = 0.041) and 49% lower MDA levels than the placebo group (10.6 ± 4.2 μM MDA). These results suggest that policosanol consumption caused a decrease in the TG and MDA content in VLDL with a smaller particle size (Figure 1 and Table 3).

### 2.5. LDL Particle Observation and Composition Analysis

TEM showed a 5% increase in LDL particle size (531 ± 8 nm^2^, *p* = 0.030) in the policosanol group at week 12, with a more distinct particle shape and morphology (Figure 3). On the other hand, the placebo group showed a 4% decrease in particle size (492 ± 10 nm^2^, *p* = 0.272) with a similar particle morphology at week 12 compared with week 0. The LDL particle diameter at week 12 in the policosanol group was 6% higher than at week 0, while the placebo group did not show a notable change (Table 3).

As shown in Table 3, the extent of LDL glycation was 12% lower in the policosanol group at 12 weeks, but the change was not significant (*p* = 0.082), while the placebo group showed an even 3% higher extent of glycation at week 12 than week 0. The policosanol group showed a 12% higher TC content and 9% lower TG content than at week 0, but the change was not significant. In contrast, the placebo group at week 12 showed a 20% lower TC level and a 2% higher TG level than at week 0. These results suggest that policosanol consumption caused an increase in the LDL particle size and cholesterol content with a decrease in the TG content, glycation extent, and oxidation extent.

### 2.6. Electromobility of LDL and Oxidation Extent

A comparison of the LDL electromobility under the nondenatured state (Figure 4A) showed that the LDL from the policosanol group at week 12 (lane 1) showed slower electromobility than week 0 (lane 2) with a stronger band intensity. In contrast, the control group at week 12 (lane 4) showed a faster electromobility and larger smear band intensity than at week 0. Native LDL (lane 5) showed the strongest band intensity with the slowest electromobility, whereas oxidized LDL (lane 6) showed the weakest band intensity with the fastest electromobility to the bottom of the gel. The more oxidized LDL exhibited faster electromobility to the bottom of the gel due to apo-B fragmentation and an increased negative charge in LDL. Native LDL, which was purified from a young and healthy control, showed 0.3 μM MDA. In contrast, the oxidized LDL by the cupric ion treatment (final 1 μM) showed the highest MDA level, around 1.3 μM, as shown in Figure 4B. The policosanol group at week 12 showed a 38% decrease in oxidation extent than at week 0 (*p* = 0.004), while the placebo group showed no significant change in the oxidation extent.

### 2.7. Change in apoA-I Contents in HDL_2_ and HDL_3_

SDS-PAGE analysis of HDL_2_ (2 mg/mL) and HDL_3_ (2 mg/mL) showed that the policosanol group at week 12 exhibited higher apoA-I expression, which increased in an intensity-dependent manner: 1.22-fold and 1.15-fold higher band intensities of apoA-I in HDL_2_ (lane 2) and HDL_3_ (lane 6), respectively, than at week 0, as shown in Figure 5A. The placebo group at week 12 showed a decrease in the apoA-I band intensities (lanes 4 and 8): 18% and 55% lower than at week 0. The increase in apoA-I in the policosanol group showed a good agreement with the increase in serum HDL-C (mg/dL) and HDL-C/TC (%), as shown in Table 2. Repeated measures ANOVA with serum HDL-C (*p* < 0.001, Figure 5B) and HDL-C/TC (%) (*p* = 0.033, Figure 5C) revealed a significant difference in the point of time and group interaction between the policosanol and the placebo groups. During the 12 weeks, the policosanol group showed a gradual increase in HDL-C (*p* < 0.001) and HDL-C/TC (%) (*p* = 0.003) with significance according to ANCOVA from the baseline, as shown in Figure 5B and Figure 5C, respectively.

### 2.8. Paraoxonase Activities in HDL_2_ and HDL_3_

As shown in Figure 6, the HDL-associated paraoxonase (PON-1) assay at the same protein concentration (2 mg/mL) showed that the policosanol group at week 12 showed the highest PON-1 activity in both HDL_2_ and HDL_3_, approximately 59 and 49 μU/L/min, while the HDL_2_ and HDL_3_ at week 0 was approximately 38–42 μU/L/min. The policosanol group showed a 1.4-fold higher HDL_2_ level at week 12 than at week 0 (*p* = 0.003), whereas the placebo group showed no change between weeks 0 and 12 (Figure 6A). The policosanol group showed a 1.3-fold higher PON-1 level of HDL_3_ at week 12 than at week 0 (*p* = 0.008), while the placebo group showed no change between weeks 0 and 12 (Figure 6B). These results suggest that policosanol consumption is linked with the specific elevation of the paraoxonase activity in both HDL_2_ and HDL_3_.

### 2.9. Ferric Ion Reduction Ability of HDL_2_ and HDL_3_

At the same protein concentration (2 mg/mL), there was no difference in the ferric ion reduction ability (FRA) for HDL_2_ between weeks 0 and 12 in the policosanol and placebo group: ~63–70 μM of ferrous ion equivalents (Figure 7A). On the other hand, the policosanol group at week 12 showed a 1.6-fold higher HDL_3_ level than at week 0 (*p* = 0.043), whereas the placebo group showed no change between weeks 0 and 12: 41–43 μM of ferrous ion equivalents, as shown in Figure 7B. These results suggest that policosanol consumption is linked with the specific elevation of HDL_3_ associated with the ferric ion reduction ability.

### 2.10. Embryo Survivability

As shown in Figure 8A, microinjection of CML (final 500 ng) into zebrafish embryos resulted in acute embryo death with up to 29 ± 3% survivability, while the PBS-alone injected embryo showed 81 ± 3% survivability. In the presence of CML, an injection of HDL_2_ from the policosanol group at week 12 resulted in the highest survivability of the injected embryo, ~75 ± 4% (*p* < 0.005 versus CML alone), whereas HDL_2_ from the policosanol group at week 0 showed 55 ± 4% survivability. On the other hand, the HDL_2_ from the placebo-injected embryo group showed similar survivability at week 0 or 12, 50–54% after 24 h post-injection. The HDL_2_ from the placebo group (weeks 0 and 12) and the policosanol group at week 0 showed protective activity against the CML-induced inflammatory death (*p* < 0.05 vs. CML alone), indicating that all the HDL showed a good anti-inflammatory activity. After 12 weeks of consumption, the HDL_2_ in the policosanol group showed a 1.5-fold higher survivability than the placebo group.

As shown in Figure 8B, the CML alone injected embryo (photo b) showed acute death, as indicated by the red arrowhead, and severe defect of development morphology, as indicated by the blue arrowhead with the slowest eye pigmentation and tail elongation. On the other hand, a co-injection of HDL_2_ ameliorated the acute death. Particularly. in HDL_2_ from the policosanol group at week 12, the most normal developmental speed and morphology was shown. Acridine orange staining showed that CML-alone injected embryos with 3.2-fold higher green fluorescence (Figure 8C), indicating that the CML injection caused apoptosis in embryos. In contrast, co-injection of HDL_2_ from the policosanol group at week 12 decreased the apoptosis by up to 52% reduction than the CML alone group. The CML-alone injected embryo also showed 3.4-fold higher red fluorescence than the PBS-alone group from the DHE staining, indicating that the apoptosis was accompanied by ROS production. A co-injection of HDL_2_ from the policosanol group at week 12 decreased the ROS amount by 63% compared with the CML-alone group. These results suggest that policosanol consumption enhanced the antioxidant and anti-inflammatory activity in HDL to result in an elevation of embryo survivability in the presence of CML.

## 3. Discussion

Cuban policosanol supplementation in Korean participants increased the serum HDL-C level and enhanced the HDL functionality to inhibit the oxidation and glycation of LDL and HDL [18]. The consumption of policosanol for 8 weeks by healthy female subjects with pre-hypertension resulted in a lower blood pressure and CETP ability by elevating the HDL/apoA-I contents and enhancing the HDL functionalities, including cholesterol efflux and insulin secretion [19]. Twelve weeks of Cuban policosanol consumption (10 and 20 mg) was associated with improved blood pressure and lipid/lipoprotein profile, such as lowering TC and LDL-C with increasing HDL-C [12,18,19]. The HDL particle quality and functionality were also enhanced by encapsulation into reconstituted HDL [16] by reducing hepatic inflammation in hyperlipidemic zebrafish [22] and spontaneously hypertensive rats [23]. In human studies, the increases in HDL-C quantity, HDL quality, and functionality were enhanced by policosanol consumption for 8 [17,19], 12 [12], and 24 weeks [18].

Many studies were carried out with different policosanol doses in different countries and ethnic populations, such as Caucasian [35,36], Cuban [37,38], and Chinese [39,40], to result in a variety of efficacies [13,41]. Interestingly, contradictory reports on the efficacy of policosanol depend on the policosanol origin and study design with healthy subjects or patients. Berthold’s group showed that policosanol is a promising phytochemical alternative for lipid reduction [35]. On the other hand, the same group later reported that policosanol consumption had no lipid-lowering effect during a 12-week study in 143 hyperlipidemic patients [36]. At the baseline of the study, however, all the participants (mean age: 56 ± 12 years old, body mass index = 27.2 ± 3.6) were patients with hypercholesterolemia who quit statin 6 weeks ago. They showed uncontrollably high serum LDL-C and TC of approximately 187 ± 36 mg/dL and 282 ± 42 mg/dL, respectively, with an additional one or two more risk factors, such as hypertension, dyslipidemia, obesity, and cigarette smoking. The patients showed lower HDL-C levels, less than 35 mg/dL, with current cigarette smoking, more than 10 cigarettes per day. In contrast, all participants in the current study were healthy subjects (52.8 ± 1.2 years old, BMI = 22.1 ± 0.4) with normotension and a normal serum lipid level around 220 ± 3 mg/dL of TC, 138 ± 4 mg/dL of LDL-C, and 64 ± 2 mg/dL of HDL-C at the baseline.

On the other hand, with the exception of Cuban policosanol [12,16,17,18], there is insufficient information on lipoprotein metabolism regarding its physiological effects, particularly in HDL functionality. Moreover, no clinical study of Cuban policosanol with Japanese populations has been conducted to test the improved blood pressure and lipid profile in healthy subjects and adverse effects. To the best of the authors’ knowledge, this study is the first to show that Cuban policosanol has efficacy in middle-aged healthy Japanese participants, who had borderline TC and LDL-C levels but high HDL-C levels, to improve dyslipidemia and blood pressure by increasing the HDL-C quantity and enhancing the HDL quality without adverse effects.

After 12 weeks of Cuban policosanol consumption, the systolic and diastolic BP were reduced significantly but still in the normal range of up to 7% (*p* < 0.001 vs. week 0) and 4% (*p* = 0.034) with a significant decrease in glycated hemoglobin from the initial level. Elevated glycated hemoglobin (HbA_1c_) levels were associated with a high risk of hypertension with lowered serum HDL-C [42]. The glycation of blood proteins, such as hemoglobin, albumin, LDL, and HDL, is linked to the exacerbation of hypertension via oxidative stress and an inflammatory process of advanced glycation end products to form atherosclerotic plaque and cause aortic stiffness. An reconstituted HDL (rHDL) containing policosanol exhibited inhibitory activity on in vitro fructose-mediated glycation [16,43]. Short-term (8 weeks) or long-term (24 weeks) policosanol consumption resulted in a decrease in the in vivo glycation in HDL and LDL [17,18]. In the same context, a meta-analysis also showed that policosanol supplementation improved hypertension and dyslipidemia [13,44]. The policosanol group at week 12 showed a remarkably lower extent of glycation and oxidation in VLDL and LDL, while the placebo group showed an increase in glycation extent after 12 weeks of consumption (Table 3). These decreases in VLDL/LDL glycation are linked with the decrease in HbA1c in the policosanol group at week 12 (Table 1) and the smear band intensity of LDL in the policosanol group at week 0 and placebo group at week 12, as shown in Figure 4 (lanes 1 and 4).

The hepatic functions were also improved remarkably by the policosanol consumption with lower serum AST, ALT, γ-GTP, and BUN levels (Table 1), but the mechanism is still unclear. The accumulation of the CML and CML-related inflammatory response in steatotic livers may play an essential role in hepatic steatosis and the pathogenesis of non-alcoholic fatty liver disease [45,46]. The nonenzymatic glycation of proteins is associated with the production of advanced glycation end products (AGE), such as CML, which is proinflammatory and toxic to hepatocytes. Therefore, inhibiting glycation by policosanol consumption may help in improving the hepatic functions.

Adding CML to HDL for 48 h increased the production of yellowish glycated fluorescence with proteolytic degradation of apoA-I and the loss of paraoxonase activity [47]. In the current study, however, consumption of 20 mg of policosanol in Japanese participants caused an elevation of apoA-I expression (Table 2), which concurs with a previous report that apoA-I expression increased and multimerization decreased in Korean participants [18]. In the current study, the significantly reduced extent of glycation in hemoglobin and VLDL/LDL agreed with previous reports [16,43] showing that Cuban policosanol-encapsulated rHDL exhibited potent anti-glycation activity against fructose-mediated glycation of HDL and apoA-I. Recently, an rHDL containing Cuban policosanol exerted anti-glycation activity to prevent proteolytic degradation of HDL and apoA-I, while rHDL containing Chinese policosanol did not show anti-glycation activity [48].

The CML also exhibited neurotoxicity and embryotoxicity in adult zebrafish and its embryo [47,48]. A microinjection of CML (final 500 ng) into zebrafish embryos caused acute embryo death, severe developmental defects, and the slowest developmental speed at 24 h post-injection [47,48]. In the current study, a co-injection of HDL_2_ from the policosanol group at week 12 showed the highest protection ability against embryonic death by neutralizing CML toxicity (Figure 8). In contrast, the HDL_2_ from the placebo group had adequate protection ability. To the best of the authors’ knowledge, these findings are the first to show that a co-injection of higher-quality HDL can neutralize CML toxicity. The enhanced HDL quality from policosanol consumption induced the highest survivability (~75 ± 4%) by neutralizing CML toxicity. Nevertheless, the other HDL showed sufficient protection ability with 50–54% survivability compared with 29 ± 3% with CML alone.

Regarding the mechanism of action, policosanol consumption resulted in an increase in HDL-C and apoA-I content in HDL via CETP inhibition, as reported elsewhere [16,17]. Cuban policosanol consumption (10–20 mg/day) resulted in a remarkable decrease in LDL oxidation and HDL glycation in healthy subjects with pre-hypertension [18]. In vitro studies showed that rHDL containing Cuban policosanol exerted potent antioxidant, anti-glycation, and anti-inflammatory activities [19,43,48] with cholesterol efflux ability [19]. The binding ability of policosanol with apoA-I for discoidal rHDL formation is crucial for exerting physiological activities by maximizing the pluripotent functionality of HDL to prevent atherosclerosis, dyslipidemia, hypertension, and dementia [49]. Since policosanol consists of long-chain aliphatic alcohols, which are extremely hydrophobic, each chain of the long-chain alcohols should be incorporated with a vesicle, such as a lipoprotein, after intake. An rHDL containing Cuban policosanol (Raydel^®^) showed a larger particle size and more particle numbers than other rHDLs containing Chinese policosanol. The rHDL-containing Cuban policosanol displayed potent anti-glycation activity to protect apoA-I and antioxidant activity to protect LDL, as reported recently [48]. The in vitro potentials of policosanol to enhance HDL functionality are linked with the in vivo efficacy in a human clinical study to improve blood pressure and dyslipidemia [12].

As summarized in Figure 9, policosanol (Raydel^®^) consumption 20 mg/day caused the simultaneous inhibition of glycation in hemoglobin (Table 1), improved the hepatic functions (Table 1), and improved the lipid profiles by raising the HDL-C levels and lowering the LDL-C/HDL-C ratio (Table 2). Consequently, improved HDL antioxidant ability (Figure 6 and Figure 7) and anti-inflammatory ability (Figure 8) were associated with a decrease in oxidation, glycation, and slower electromobility of VLDL and LDL (Table 3, Figure 2 and Figure 4).

A limitation of this study was that the data on lifestyle, such as diet consumption, smoking, and alcohol drinking frequency, exercise intensity, and time were obtained from self-reported questionnaires. The validation of these data for diet, smoking, drinking, and exercise might be intricate for distinguishing the consumption of policosanol or placebo effect. In addition, methodological and practical difficulties due to space limitation in rotor for ultracentrifugation were related with the selection of participants. To achieve more reliable and accurate results, the four steps of ultracentrifugation for a total of 96 h and subsequent dialysis for each 24 h, from VLDL, LDL, HDL_2_, to HDL_3_ should be carried out in the same batch, simultaneously. Antioxidant abilities, PON-1 and FRA, and anti-inflammatory activities are very sensitive to change during the purification step. However, since the rotor contains a maximum of 44 holes, we had to select blood samples of 15 samples from the policosanol group, 17 samples from the placebo group, 10 samples from the young control group for the simultaneous ultracentrifugation. These low sample sizes (n = 32) for lipoprotein analysis could be a limitation of the current study. Another concern was the unequal distribution of menopausal women between the groups; the policosanol (Female n = 7) and placebo group (Female n = 8) contained four and seven post-menopausal women, respectively. These unequal distributions of menopausal status between groups might interfere with the interpretation of the current results since menopausal women displayed more atherogenic lipid and lipoprotein profiles with increased dysfunctional HDL [35]. In a future study, the detailed properties of HDL, particle size, and compositions from each participant should be investigated to observe trends in different parameters between the groups. Therefore, it will be important to know which parameters are more influential to the HDL quality and functionality in vivo across the participants by the policosanol consumption.

In conclusion, 12 weeks of Cuban policosanol (Raydel^®^) consumption resulted in a significant decrease in BP and glycated hemoglobin with improvements of the hepatic parameters via lowered oxidation and glycation of VLDL and LDL through enhanced HDL functionalities with a higher apoA-I content. Policosanol consumption enhanced the HDL functionalities on the in vitro antioxidant abilities (PON and FRA) and anti-inflammatory activities in the zebrafish embryos to protect against acute death in the presence of CML, a proinflammatory neurotoxin.

## 4. Materials and Methods

### 4.1. Policosanol

Raydel^®^ policosanol tablet (two tablets of 10 mg, total 20 mg per day) was obtained from Raydel Japan (Tokyo, Japan), which was manufactured with Cuban policosanol at Raydel Australia (Thornleigh, Sydney). Cuban policosanol was defined as genuine policosanol with a specific ratio of each ingredient [50]: 1-Tetracosanol (C_24_H_49_OH, 0.1–20 mg/g); 1-hexacosanol (C_26_H_53_OH, 30.0–100.0 mg/g); 1-heptacosanol (C_27_H_55_OH, 1.0–30.0 mg/g); 1-octacosanol (C_28_H_57_OH, 600.0–700.0 mg/g); 1-nonacosanol (C_29_H_59_OH, 1.0–20.0 mg/g); 1-triacontanol (C_30_H_61_OH, 100.0–150.0 mg/g); 1-dotriacontanol (C_32_H_65_OH, 50.0–100.0 mg/g); 1-tetratriacontanol (C_34_H_69_OH, 1.0–50.0 mg/g).

### 4.2. Participants

Healthy male and female volunteers with normal lipid levels and normal blood pressure were recruited nationwide in Japan via newspaper and internet advertisements between September 2021 and May 2022. The inclusion criteria were LDL-C levels in the normal range (120–160 mg/dL) and age between 20 and 65 years old. The exclusion criteria were as follows: (1) Maintenance treatment for metabolic disorder, including dyslipidemia, hypertension, and diabetes; (2) severe hepatic, renal, cardiac, respiratory, endocrinological, and metabolic disorder disease; (3) allergies; (4) heavy drinkers, more than 30 g of alcohol per day; (5) taking medicine or functional food products that may affect the lipid metabolism, including raising HDL-C or lowering LDL-C concentration, and lowering triglyceride concentration; (6) current and past smoker; (7) women in pregnancy, lactation, or planning to become pregnant during the study period; (8) person who had more than 200 mL of blood donation within 1 month or 400 mL of blood within 3 months before starting this clinical trial; (9) a person who participated in other clinical trials within the last 3 months or currently is participating in another clinical trial; (10) those who consumed more 2000 kcal per day; (11) others considered unsuitable for the study at the discretion of the principal investigator. The study was approved by the Koseikai Fukuda Internal Medicine Clinic (Osaka, Japan, IRB approval number 15000074, approval date on 18 September 2021).

### 4.3. Study Design

This study was a double-blinded, randomized, and placebo-controlled trial with a 12-week treatment period. After an initial screening, 72 subjects (Male 36, Female 36) with 120 mg/dL ≤ LDL <160 mg/dL were selected as shown in Figure 10.

After allocating the participants into two groups, they were directed to take two tablets per day containing policosanol 10 mg (Raydel^®^) or a placebo. The tablet for the policosanol group included policosanol (10 mg), hydroxypropyl cellulose, carboxymethyl cellulose, maltodextrin, lactose, and crystalline cellulose. The tablet for the placebo group contained maltodextrin (10 mg) rather than policosanol.

All participants received advice to avoid excess food (<1800 and 1500 kcal for men and women, respectively, per day), cholesterol (<600 mg per day), and alcohol drinking (<30 and <15 g of ethanol for men and women, respectively, per day), and no smoking both direct and indirect, which can interfere with the lipoprotein metabolism. They were also instructed to avoid intense exercise (<30 min daily at 60–80% maximum capacity). After 12 weeks of consumption, the blood parameters of all participants who completed the program were analyzed. Then, the lipid and lipoprotein parameters were analyzed after excluding those who violated dietary and exercise guidelines, such as omitting daily consumption, overeating, a significantly more fat diet, smoking, and heavy drinking, and failed the other exclusion criteria after stratified analysis based on the self-reported questionnaire.

### 4.4. Anthropometric Analysis

The blood pressure was measured using an Omron HEM-907 (Kyoto, Japan) with a total of three times of measurements, and the average was recorded. The height, body weight, and body mass index (BMI) were measured individually using a DST-210N (Muratec KDS Co., Ltd., Kyoto, Japan).

### 4.5. Blood Analysis

After fasting overnight, blood samples were collected in ethylenediaminetetraacetic acid (EDTA)-coated tubes and centrifuged at 3000× *g* for 15 min at 4 °C for the plasma assays. The samples were subjected to 19 blood biochemical assays by BML Inc. (Tokyo, Japan): Total protein, albumin, albumin and globulin ratio, aspartate transferase (AST), alanine aminotransferase (ALT), gamma-glutamyl transpeptidase (γ-GTP), creatinine, glucose, uric acid, blood urea nitrogen (BUN), lactate dehydrogenase (LDH), total bilirubin, glycated hemoglobin (hemoglobin A_1c_, HbA_1c_), high sensitivity C-reactive protein (hsCRP), total cholesterol (TC), triglyceride (TG), high-density lipoprotein cholesterol (HDL-C), low-density lipoprotein cholesterol (LDL-C), apolipoprotein B (apo-B), and apolipoprotein A-I (apoA-I).

### 4.6. Isolation of Lipoproteins and Quantification

Very low-density lipoproteins (VLDL, d < 1.019 g/mL), LDL (1.019 < d < 1.063), HDL_2_ (1.063 < d < 1.125), and HDL_3_ (1.125 < d < 1.225) were isolated from individual subject sera via sequential ultracentrifugation for 96 h [51], with the density adjusted by adding NaCl and NaBr according to standard protocols [52]. As a control serum for native VLDL and LDL, the blood from young and healthy human males (n = 10, mean age, 23 ± 2 years old) was donated voluntarily after fasting overnight. The samples were centrifuged for 24 h for each lipoprotein fraction at 10 °C at 100,000× g using an Himac NX (Hitachi, Tokyo, Japan) equipped with a fixed angle rotor P50AT4-0124 at the Raydel Research Institute (Daegu, Korea). After centrifugation, each lipoprotein sample was dialyzed extensively against Tris-buffered saline (TBS; 10 mM Tris-HCl, 140 mM NaCl, and 5 mM ethylene-diamine-tetraacetic acid (EDTA) [pH 8.0]) for 24 h to remove the NaBr.

For each lipoprotein purified individually, the total cholesterol (TC) and TG levels were measured using commercially available kits (cholesterol, T-CHO, and TG, Cleantech TS-S; Wako Pure Chemical, Osaka, Japan). The protein concentrations of the lipoproteins were determined using a Lowry protein assay, as modified by Markwell et al. [53], using the Bradford assay reagent (Bio-Rad, Seoul, South Korea) with bovine serum albumin (BSA) as the standard.

### 4.7. Quantification of Oxidation Extent in VLDL and LDL

The degree of oxidation of the individual VLDL (0.5 mg/mL of protein) and LDL (1.0 mg/mL of protein) was assessed by measuring the concentration of oxidized species in the lipoproteins using the thiobarbituric acid reactive substances (TBARS) method with malondialdehyde (MDA) as a standard [54]. The relative electrophoretic mobility depends on the intact charge and three-dimensional structure of VLDL and LDL.

### 4.8. Oxidation of VLDL and LDL

Oxidized VLDL (oxVLDL) and LDL (oxLDL) were produced by incubating the native VLDL (0.5 mg/mL of protein) or LDL fraction (1.0 mg/mL of protein), which was purified from young and healthy males, with CuSO_4_ (Sigma # 451657) at a final concentration of 10 and 1 μM for VLDL and LDL, respectively, for 4 h at 37 °C. The oxVLDL and oxLDL were then filtered (0.22 μm filter) and analyzed using a thiobarbituric acid reactive substance (TBARS) assay to determine the extent of oxidation with malondialdehyde (MDA, Sigma # 63287) standard, as described elsewhere [54].

### 4.9. Agarose Electrophoresis

The relative electromobility of the VLDL and LDL (5 μg of protein) was compared under a non-natured state on 0.5% agarose gel (120 mm length × 60 mm width × 5 mm thickness). The electrophoresis was carried out with 50 V for 1 h in Tris-acetate-EDTA buffer (pH 8.0), as described previously [55]. The apo-B in VLDL and LDL were visualized by Coomassie brilliant blue staining (final 1.25%). More oxidized VLDL and LDL were moved faster to the bottom of the gel due to apo-B fragmentation and the increase in negative charge.

### 4.10. Electron Microscopy

Transmission electron microscopy (TEM, Hitachi H-7800; Ibaraki, Japan) at the Raydel Research Institute (Daegu, Korea) was performed at an acceleration voltage of 80 kV. VLDL and LDL were stained negatively with 1% sodium phosphotungstate (PTA; pH 7.4) with a final apolipoprotein concentration of 0.3 mg/mL in TBS. Five μL of the lipoprotein suspension was blotted with filter paper and replaced immediately with a 5 μL droplet of 1% PTA. After a few seconds, the stained lipoprotein fraction was blotted onto a Formvar carbon-coated 300 mesh copper grid and air-dried. The shape and size of the LDL were determined by TEM at 40,000× magnification, according to a previous report [56].

### 4.11. Paraoxonase Assay

The paraoxonase-1 (PON-1) activity in HDL_2_ and HDL_3_ toward paraoxon was determined by evaluating the hydrolysis of paraoxon into *p*-nitrophenol and diethylphosphate, which was catalyzed by the enzyme [57]. The PON-1 activity was then determined by measuring the initial velocity of *p*-nitrophenol production at 37 °C, as determined by measuring the absorbance at 415 nm (microplate reader, Bio-Rad model 680; Bio-Rad, Hercules, CA, USA).

### 4.12. Ferric Ion Reduction Ability Assay

The ferric ion reduction ability (FRA) was determined using the method reported by Benzie and Strain [58]. Briefly, the FRA reagents were freshly prepared by mixing 20 mL of 0.2 M acetate buffer (pH 3.6), 2.5 mL of 10 mM 2,4,6-tripyridyl-S-triazine (Fluka Chemicals, Buchs, Switzerland), and 2.5 mL of 20 mM FeCl_3_∙6H_2_O. The antioxidant activities of HDL (2 mg/mL) were estimated by measuring the increase in absorbance induced by the ferrous ions generated. Freshly prepared FRA reagent (300 μL) was mixed with HDL_2_ (2 mg/mL) and HDL_3_ (2 mg/mL) as an antioxidant source. The FRA was determined by measuring the absorbance at 593 nm every 2 min over a 60 min period at 25 °C using a UV-2600i spectrophotometer.

### 4.13. Electrophoretic Patterns of HDL

The relative compositions of the apolipoproteins and band intensity of apoA-I in HDL_2_ and HDL_3_ were compared using 12% SDS-PAGE in the denatured state. The gels were then stained with 0.125% Coomassie Brilliant Blue, after which the relative band intensities were compared by band scanning using Gel Doc^®^ XR (Bio-Rad) with Quantity One software (version 4.5.2) and Image J software (http://rsb.info.nih.gov/ij/, accessed on 15 December 2022).

### 4.14. Data Analysis

All analyses in Table 1, Table 2 and Table 3 were normalized using a homogeneity test of the variances through Levene’s statistics. Nonparametric statistics were performed using the Kruskal–Wallis test if not normalized. For Table 1, comparisons between the policosanol and placebo with respect to all assessments were analyzed using an analysis of covariance (ANCOVA) with the independent variable as the baseline and treatment. For Table 2, repeated measure ANOVA was used for comparison of the score changes between the two groups during the same period. The differences in the placebo or policosanol group over the follow-up time were analyzed to compare the point of time and group interaction. For Table 3, significant changes between the baseline and follow-up values within groups were assessed using a paired *t*-test. Statistical power was estimated using the program G*Power 3.1.9.7 (G*Power from University of Düsseldorf, Düsseldorf, Germany). All tests were two-tailed, and the statistical significance was *p* < 0.05. Data were analyzed using the SPSS software version 29.0 (IBM, Chicago, IL, USA).

## Figures and Tables

**Figure 1 ijms-24-05185-f001:**
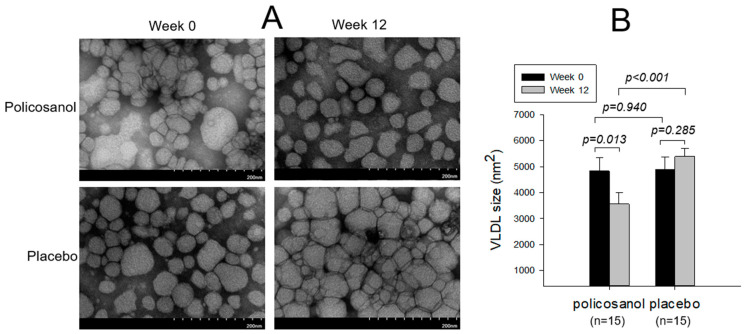
TEM image of VLDL (**A**) at a magnification with 40,000× and area analysis (**B**) of VLDL from each group between weeks 0 and 12. VLDL particle size in each group between weeks 0 and 12 was compared by the paired *t*-test. One graduation of the scale bar indicates 20 nm.

**Figure 2 ijms-24-05185-f002:**
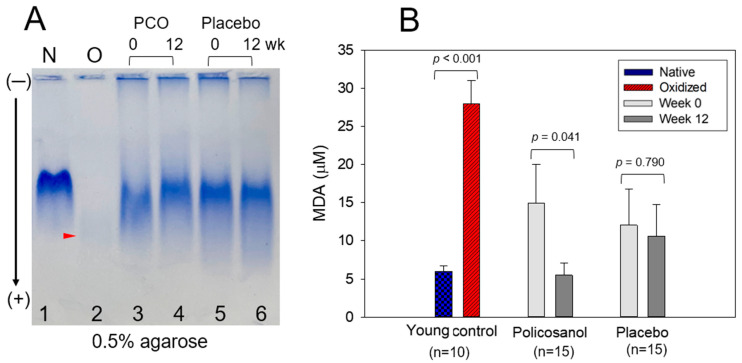
Comparison of the electromobility (**A**) and quantification of oxidation extent (**B**) in VLDL from the policosanol (PCO, 20 mg) and placebo groups. (**A**) Electrophoresis of VLDL on 0.5% agarose gel (120 mm length × 60 mm width × 5 mm thickness) under a nondenatured state. The electrophoresis was carried out with 50 V for 1 h in Tris-acetate-ethylene-diamine-tetraacetic acid buffer (pH 8.0). The apo-B in VLDL was visualized by Coomassie brilliant blue staining (final 1.25%). Lane N, native VLDL (0.5 mg/mL of protein), which was purified from young and healthy control; lane O, oxidized VLDL, cupric ion (final 10 μM) treated for 4 h; PCO, policosanol. The red arrowhead indicates an oxidized VLDL band. (**B**) Quantification of the oxidized VLDL contents by a thiobarbituric acid reactive substance assay using a malondialdehyde (MDA) standard. The data are expressed as the mean ± SEM from three independent experiments with duplicate samples. In young control, cupric ion (final 10 μM) treated VLDL (O) was compared with native VLDL alone (N) by a paired *t*-test. The extent of oxidation in each group between weeks 0 and 12 was compared using a paired *t*-test. N, native VLDL from young and healthy male volunteers; O, cupric ion treated VLDL (oxidized VLDL).

**Figure 3 ijms-24-05185-f003:**
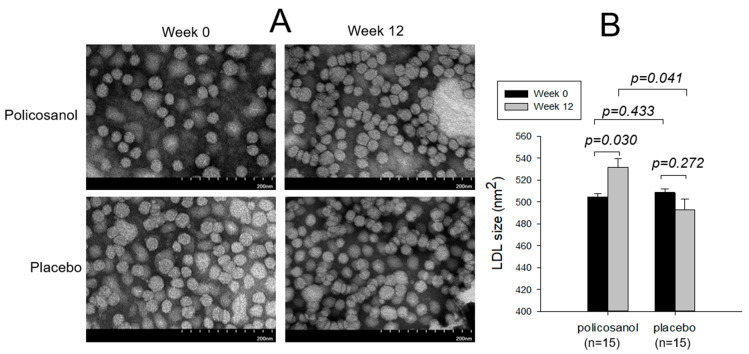
TEM image of LDL (**A**) at a magnification with 40,000× and area analysis (**B**) of LDL from each group between weeks 0 and 12. LDL particle size in each group between weeks 0 and 12 was compared by a paired *t*-test. One graduation of the scale bar indicates 20 nm.

**Figure 4 ijms-24-05185-f004:**
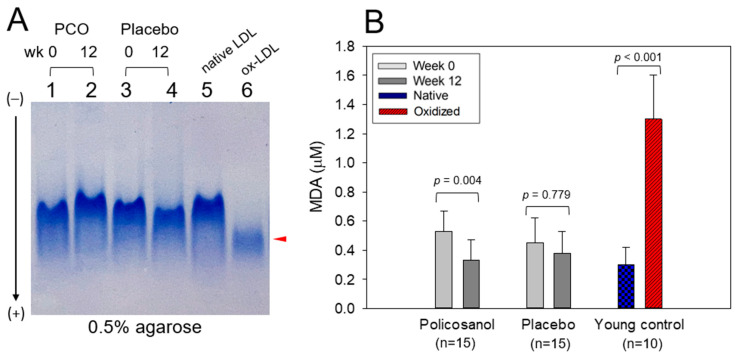
Comparison of the electromobility (**A**) and quantification of oxidation extent (**B**) in LDL from the policosanol (PCO, 20 mg) group and placebo group between weeks 0 and 12. (**A**) Electrophoresis under the nondenatured state on 0.5% agarose gel (120 mm length × 60 mm width × 5 mm thickness). Electrophoresis was carried out with 50 V for 1 h in Tris-acetate-ethylene-diamine-tetraacetic acid buffer (pH 8.0). The red arrowhead indicates an oxidized LDL band. The apo-B in LDL was visualized by Coomassie brilliant blue staining (final 1.25%). Lane N, native LDL (1.0 mg/mL of protein), which was purified from a young and healthy control. Lane O, oxidized LDL, cupric ion (final 1 μM) treated for 4 h; PCO, policosanol. (**B**) Quantification of the oxidized LDL contents by a thiobarbituric acid reactive substance assay using a malondialdehyde (MDA) standard. The data are expressed as mean ± SEM from three independent experiments with duplicate samples. In the young control, cupric ion (final 10 μM) treated LDL (O) was compared with native LDL alone (N) using a paired *t*-test. The oxidation extent in each group between weeks 0 and 12 was compared by the paired *t*-test.

**Figure 5 ijms-24-05185-f005:**
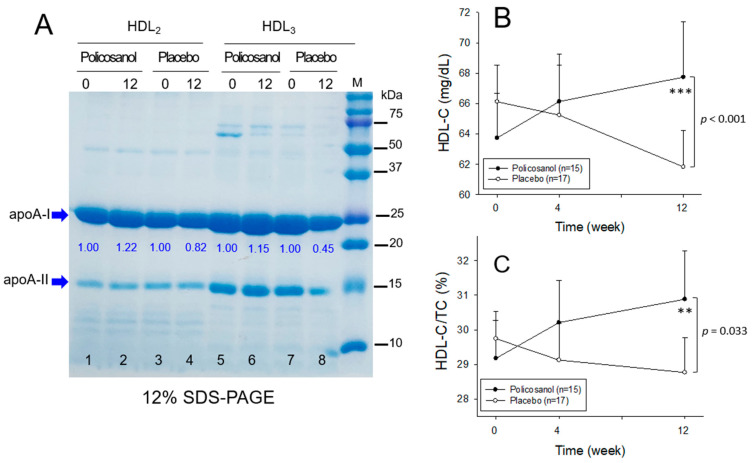
Changes in the apoA-I contents in HDL, HDL-C, and %HDL-C in TC during 12 weeks of consumption of policosanol and placebo. HDL-C, high-density lipoproteins-cholesterol; TC, total cholesterol. (**A**) Electrophoretic patterns of apolipoproteins in HDL in the denatured state from the participants in the policosanol (n = 15) and placebo group (n = 15). The numbers in blue font below the apoA-I band indicate band intensity between weeks 0 and 12. The bands were visualized by 0.125% Coomassie Blue staining and quantified by band scanning using Gel Doc^®^ XR (Bio-Rad) with Quantity One software (version 4.5.2)) and Image J software (http://rsb.info.nih.gov/ij/, accessed on 29 December 2022) from four gels of the same sample. Lane M, molecular weight standards (Bio-Rad Cat#161-0374). (**B**) Change in the HDL-C (mg/dL) during 12 weeks of consumption of policosanol (n = 15) and placebo (n = 17) from repeated measurement ANOVA. The data are expressed as the mean ± SEM. ***, *p* < 0.001 from the analysis of covariance (ANCOVA) model with the independent variable as the baseline and treatment. (**C**) Change in percentage of HDL-C in TC (%) during 12 weeks of consumption of policosanol (n = 15) and placebo (n = 17) from repeated measurement ANOVA. The data are expressed as the mean ± SEM. **, *p* < 0.01 from the analysis of covariance (ANCOVA) model with the independent variable as the baseline and treatment.

**Figure 6 ijms-24-05185-f006:**
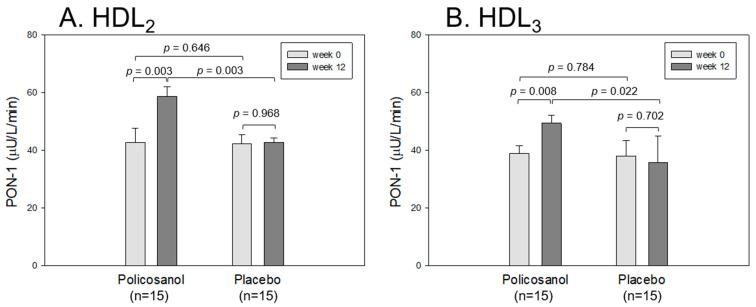
Comparison of the paraoxonase (PON-1) activity in HDL_2_ (**A**) and HDL_3_ (**B**) from the policosanol and placebo groups between weeks 0 and 12. The PON-1 activity is expressed as the initial velocity of *p*-nitrophenol production per min (μU/L/min) at 37 °C during 60 min of incubation. The data are expressed as the mean ± SD from three independent experiments with duplicate samples. The PON-1 activity in each group between weeks 0 and 12 was compared using a paired *t*-test.

**Figure 7 ijms-24-05185-f007:**
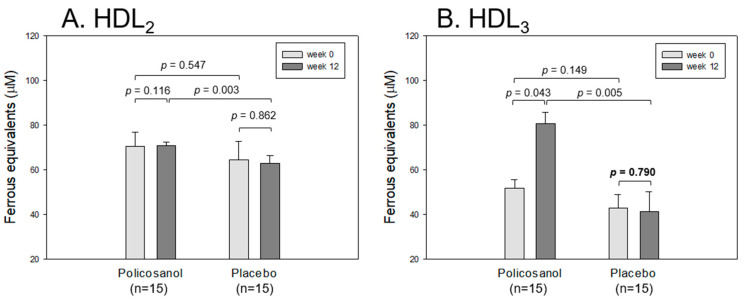
Comparison of the ferric ion reduction ability (FRA) of HDL_2_ (**A**) and HDL_3_ (**B**) in the policosanol and placebo group between weeks 0 and 12. The FRA activity was expressed as the concentration of vitamin C (μM), which is equivalent to reducing the amount of ferric ion (μM) per hour. The data are expressed as the mean ± SD from three independent experiments with duplicate samples. FRA activity in each group between weeks 0 and 12 was compared using a paired *t*-test.

**Figure 8 ijms-24-05185-f008:**
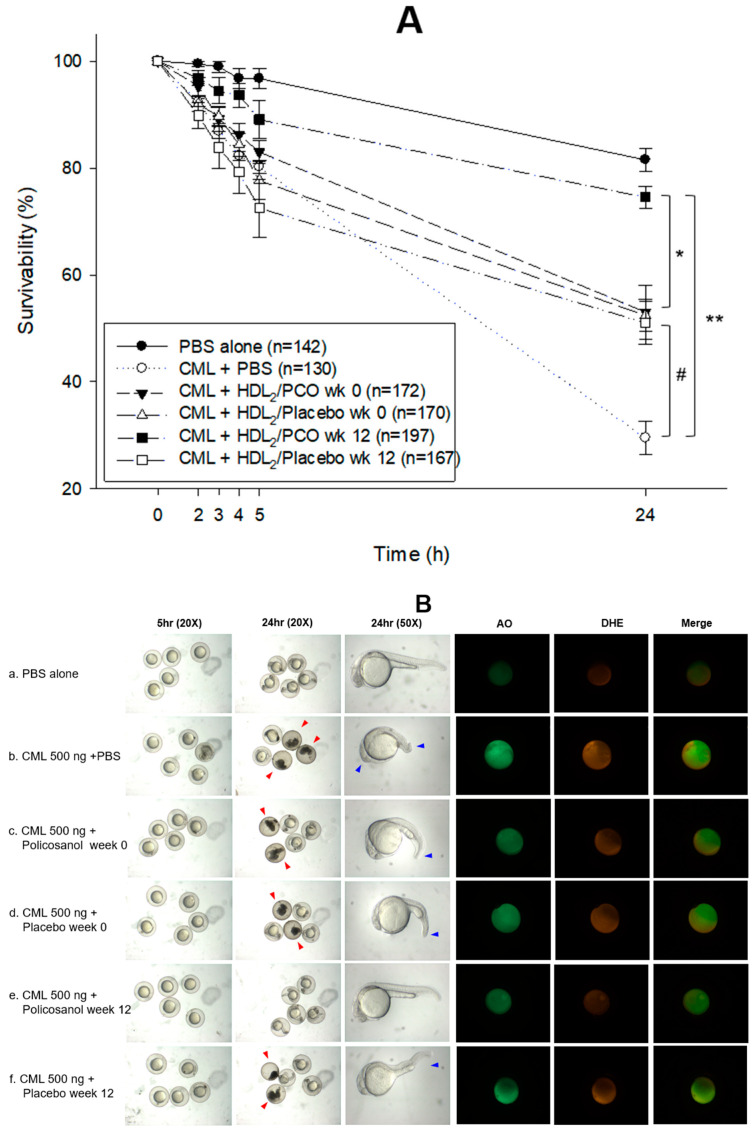
Comparison of the protective activity of HDL_2_ (1 mg/mL) from the policosanol and placebo groups against carboxymethyllysine (CML, final 500 ng) toxicity in zebrafish embryos. (**A**) Survivability of zebrafish embryos during 24 h post-injection in the presence of HDL_2_ (20 ng) and CML. Embryo numbers were adjusted from three independent experiments. The data are expressed as the mean ± SD from five independent experiments. Statistically significant differences in multiple groups were compared using a one-way analysis of variance (ANOVA). *, *p* < 0.05 between the policosanol group at weeks 0 and 12; **, *p* < 0.005 between the policosanol group at week 12 and CML alone; #, *p* < 0.05 between the placebo group at week 12 and CML alone. (**B**) Stereo image of the zebrafish embryos at 5 and 24 h post-injection. The red arrowheads indicate defective development and death of embryos in the CML group (photographs b, c, d, and f). The blue arrowhead indicates the slowest developmental speed in eye pigmentation and tail elongation in the CML group at 24 h post injection (photographs b, c, d, and f). Fluorescence image of dihydroethidium (DHE, Ex = 585 nm, Em = 615 nm) stained and acridine orange (AO, Ex = 505 nm, Em = 535 nm) stained embryo at 5 h post-injection. The extent of ROS production and apoptosis was visualized by DHE and AO staining, respectively. (**C**) Quantification of the fluorescence from DHE-stained and AO-stained embryo images using Image J software (http://rsb.info.nih.gov/ij/, accessed on 3 January 2023). The data are expressed as the mean ± SD from five independent experiments. The statistical differences of multiple groups were compared using a one-way analysis of variance (ANOVA). AU, arbitrary unit.

**Figure 9 ijms-24-05185-f009:**
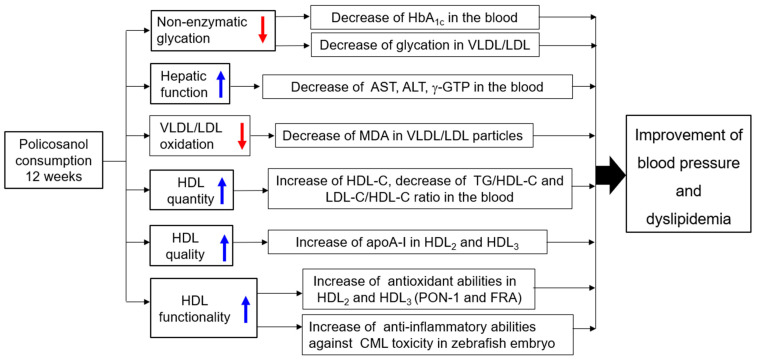
Proposed mechanism of policosanol to improve blood pressure and dyslipidemia after 12 weeks of consumption. apoA-I, apolipoprotein A-I; AST, aspartate transferase; ALT, alanine aminotransferase; CML, carboxymethyllysine; FRA, ferric ion reduction ability; γ-GTP, gamma-glutamyl transpeptidase; HbA1c, hemoglobin A1c; LDL, low-density lipoproteins; MDA, malondialdehyde; PON-1, paraoxonase-1; TC, total cholesterol; TG, triglyceride; VLDL, very low-density lipoproteins.

**Figure 10 ijms-24-05185-f010:**
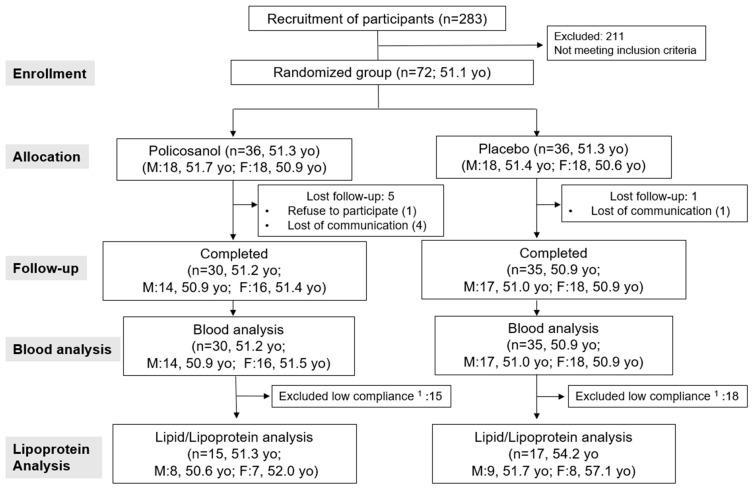
Study design and participant allocation for analysis. ^1^ The participants were excluded for the following reasons: Violated dietary and exercise guidelines, such as omitting daily consumption, overeating, a significantly more fat diet, smoking, and heavy drinking; failed the exclusion criteria after stratified analysis based on a self-reported questionnaire of daily diet and exercise.

**Table 1 ijms-24-05185-t001:** Anthropometric profiles and blood parameters between the policosanol (20 mg) and placebo groups *.

		Policosanol 20 mg (n = 30)	Week 0 vs. 12 (*p* ^†^)	Placebo (n = 35)	Week 0 vs. 12 *p* ^†^	*p* ^‡^
Age (year) (min., max.)	Week 0	51.2 ± 1.7	1.000	50.9 ± 1.7	1.000	0.915
	Week 12	51.2 ± 1.7 (38, 62)		50.9 ± 1.7 (40, 63)		
SBP (mmHg)	Week 0	114.0 ± 0.8	<0.001	115.7 ± 2.4	0.118	0.026
	Week 12	106.1 ± 2.6		112.7 ± 2.2		
DBP (mmHg)	Week 0	70.6 ± 1.8	0.034	70.8 ± 1.8	0.461	0.195
	Week 12	67.8 ± 2.12		70.0 ± 1.6		
Heart pulse rate (BPM)	Week 0	72.4 ± 1.8	0.528	70.2 ± 1.8	0.938	0.487
	Week 12	73.5 ± 2.5		70.1 ± 1.9		
Height (cm)	Week 0	163.7 ± 1.6	1.000	164.6 ± 1.5	1.000	0.093
	Week 12	163.7 ± 1.6		164.6 ± 1.5		
Body weight (kg)	Week 0	58.6 ± 1.7	0.531	60.7 ± 1.5	0.006	0.242
	Week 12	58.5 ± 1.7		60.2 ± 1.4		
BMI (kg/m^2^)	Week 0	21.8 ± 0.4	0.459	22.4 ± 0.4	0.006	0.307
	Week 12	21.8 ± 0.4		22.2 ± 0.4		
Total protein (g/dL)	Week 0	7.1 ± 0.1	0.874	7.0 ± 0.1	0.159	0.386
	Week 12	7.1 ± 0.1		7.1 ± 0.1		
Albumin (g/dL)	Week 0	4.4 ± 0.1	0.500	4.3 ± 0.0	0.456	0.448
	Week 12	4.4 ± 0.1		4.4 ± 0.0		
A/G (ratio)	Week 0	1.64 ± 0.03	0.243	1.66 ± 0.04	0.239	0.911
	Week 12	1.61 ± 0.03		1.63 ± 0.03		
AST (IU/L)	Week 0	20.8 ± 1.4	0.022	20.8 ± 0.9	0.422	0.017
	Week 12	19.0 ± 0.8		21.3 ± 1.0		
ALT (IU/L)	Week 0	21.5 ± 2.8	0.013	21.5 ± 2.4	0.971	0.032
	Week 12	17.9 ± 1.5		21.4 ± 2.5		
γ-GTP (IU/L)	Week 0	30.0 ± 4.4	0.016	27.9 ± 3.4	0.775	0.016
	Week 12	25.4 ± 3.0		28.3 ± 3.6		
Creatinine (mg/dL)	Week 0	0.73 ± 0.02	0.883	0.75 ± 0.03	0.608	0.675
	Week 12	0.73 ± 0.03		0.74 ± 0.03		
Glucose (mg/dL)	Week 0	90.0 ± 1.7	0.652	91.2 ± 1.4	0.890	0.968
	Week 12	90.5 ± 1.5		91.3 ± 1.4		
Uric acid (mg/dL)	Week 0	5.1 ± 0.2	0.891	5.2 ± 0.2	0.887	0.979
	Week 12	5.1 ± 0.2		5.2 ± 0.2		
BUN (mg/dL)	Week 0 Week 12	13.4 ± 0.4	0.052	13.5 ± 0.5	<0.001	0.001
		12.6 ± 0.6		14.6 ± 0.5		
LDH (IU/L)	Week 0	158.3 ± 4.6	0.827	165.5 ± 4.2	0.874	0.663
	Week 12	157.9 ± 4.2		165.2 ± 4.1		
Total bilirubin (mg/dL)	Week 0 Week 12	0.82 ± 0.05 0.86 ± 0.05	0.294	0.76 ± 0.04 0.85 ± 0.05	0.030	0.419
HbA_1c_ (%)	Week 0	5.5 ± 0.0	0.009	5.5 ± 0.1	0.212	0.024
	Week 12	5.3 ± 0.1		5.4 ± 0.1		
hsCRP (mg/dL)	Week 0	0.05 ± 0.02	0.422	0.11 ± 0.06	0.278	0.540
	Week 12	0.03 ± 0.78		0.05 ± 0.01		
apoA-I (mg/dL) (n = 15)	Week 0	165.5 ± 2.3	0.045	164.6 ± 5.5	0.347	0.028
	Week 12	182.8 ± 8.1		160.5 ± 2.8		
apo-B (mg/dL) (n = 15)	Week 0	97.9 ± 2.7	0.465	101.3 ± 1.8	0.333	0.182
	Week 12	100.1 ± 5.5		98.2 ± 1.5		

*, Data are expressed as the mean ± SEM (standard error of the mean). *p* ^†^, paired *t*-test performed for values obtained between weeks 0 and 12. *p*
^‡^, analysis of covariance (ANCOVA) model with the independent variable as baseline and treatment. A/G, albumin/globulin; AST, aspartate transferase; ALT, alanine aminotransferase; BUN, blood urea nitrogen; BMI, body mass index; BPM, beat per minute; DBP, diastolic blood pressure; γ-GTP, gamma-glutamyl transpeptidase; HbA1c, hemoglobin A1c; hs-CRP, high sensitivity C-reactive protein; LDH, lactate dehydrogenase; SBP, systolic blood pressure; TC, total cholesterol; TG, triglyceride.

**Table 2 ijms-24-05185-t002:** Repeated measures ANOVA of blood lipid parameters between the policosanol (PCO, 20 mg) and placebo groups *.

	Groups	Week 0	Week 4	Week 8	Week 12	Sources	F	*p* ^‡^
		Mean ± SEM	Mean ± SEM	Mean ± SEM	Mean ± SEM			
TC (mg/dL)	placebo (n = 17)	222.1 ± 5.0	222.5 ± 4.9	219.4 ± 5.2	214.8 ± 3.9	Time × group	1.407	0.246
	PCO 20 mg (n = 15)	217.9 ± 3.8	218.5 ± 4.5	212.7 ± 4.1	218.5 ± 3.7			
	*p* ^†^	0.521	0.859	0.456	0.280			
TG (mg/dL)	placebo (n = 17)	74.6 ± 6.5	84.6 ± 8.9	86.4 ± 10.3	100.2 ± 16.6	Time × group	1.586	0.215
	PCO 20 mg (n = 15)	97.7 ± 12.3	94.1 ± 8.1	125.9 ± 21.0	101.7 ± 14.9			
	*p* ^†^	0.095	0.952	0.236	0.063			
HDL-C (mg/dL)	placebo (n = 17)	66.1 ± 2.4	65.2 ± 3.3	63.8 ± 3.2	61.8 ± 2.4	Time × group	10.583	<0.001
	PCO 20 mg (n = 15)	63.7 ± 2.9	66.1 ± 3.1	63.9 ± 2.9	67.7 ± 3.6			
	*p* ^†^	0.532	0.158	0.329	<0.001			
LDL-C (mg/dL)	placebo (n = 17)	141.0 ± 3.2	140.3 ± 3.2	138.4 ± 3.0	133.0 ± 3.1	Time × group	1.944	0.128
	PCO 20 mg (n = 15)	134.5 ± 3.5	133.5 ± 3.5	123.7 ± 3.7	130.4 ± 3.6			
	*p* ^†^	0.186	0.404	0.013	0.956			
TG /HDL-C (ratio)	placebo (n = 17)	1.16 ± 0.11	1.42 ± 0.21	1.46 ± 0.2	1.73 ± 0.31	Time × group	2.571	0.074
	PCO 20 mg (n = 15)	1.66 ± 0.27	1.5 ± 0.16	2.07 ± 0.36	1.68 ± 0.31			
	*p* ^†^	0.099	0.524	0.355	0.018			
LDL-C /HDL-C (ratio)	placebo (n = 17)	2.17 ± 0.08	2.23 ± 0.11	2.23 ± 0.09	2.2 ± 0.09	Time × group	2.176	0.096
	PCO 20 mg (n = 15)	2.16 ± 0.09	2.08 ± 0.1	2.00 ± 0.12	1.99 ± 0.1			
	*p* ^†^	0.850	0.239	0.018	0.054			
HDL-C/TC (%)	placebo (n = 17)	29.7 ± 0.8	29.1 ± 1.1	28.9 ± 1.0	28.8 ± 1.0	Time × group	3.033	0.033
	PCO 20 mg (n = 15)	29.2 ± 1.1	30.2 ± 1.2	30.1 ± 1.4	30.9 ± 1.4			
	*p* ^†^	0.671	0.085	0.086	0.003			
RC (mg/dL)	placebo (n = 17)	14.9 ± 1.3	16.9 ± 1.8	17.2 ± 2.1	20.0 ± 3.3	Time × group	1.582	0.216
	PCO 20 mg (n = 15)	19.6 ± 2.4	18.9 ± 1.6	25.1 ± 4.2	20.3 ± 3.0			
	*p* ^†^	0.090	0.945	0.244	0.057			

*, Data are expressed as the mean ± SEM (standard error of the mean). Estimated statistical power is 92% based on the calculation using the program G*Power 3.1.9.7 (G*Power from University of Düsseldorf, Düsseldorf, Germany). *p* ^†^, analysis of covariance (ANCOVA) model with the independent variable as the baseline and treatment. *p*
^‡^, analysis of repeated measures ANOVA. LDL-C, direct low-density lipoprotein cholesterol; HDL-C, high-density lipoprotein cholesterol; HDL-C/TC (%), percentage of HDL-C in TC; RC, remnant cholesterol; PCO, policosanol; TC, total cholesterol; TG, triglyceride.

**Table 3 ijms-24-05185-t003:** Lipid compositions, oxidized extent, and extent of VLDL and LDL glycation between the policosanol and placebo groups.

		Policosanol 20 mg		*p*	Placebo		*p*
		Week 0 n = 15 (Mean ± SEM)	Week 12 n = 15 (Mean ± SEM)		Week 0 n = 15 (Mean ± SEM)	Week 12 n = 15 (Mean ± SEM)	
VLDL	FI (Glycated)	3851 ± 144	3572 ± 143	0.067	4074 ± 230	4379 ± 274	0.223
	MDA (μM)	14.9 ± 5.1	5.5 ± 1.60	0.041	12.1 ± 4.7	10.6 ± 4.2	0.790
	Size (nm^2^)	4827 ± 529	3562 ± 426	0.013	4882 ± 496	5389 ± 309	0.285
	Diameter (nm)	75.3 ± 3.1	65.3 ± 2.3	0.016	74.1 ± 3.7	78.8 ± 3.8	0.381
	TC (mg/dL)	50.4 ± 0.9	73.3 ± 8.5	0.134	57.0 ± 5.3	53.0 ± 5.7	0.207
	TG (mg/dL)	95.0 ± 7.2	81.2 ± 5.3	0.051	79.9 ± 13.8	84.0 ± 12.1	0.459
LDL	FI (Glycated)	4958 ± 266	4416 ± 121	0.082	4934 ± 622	5109 ± 900	0.640
	MDA (μM)	0.53 ± 0.14	0.33 ± 0.14	0.004	0.45 ± 0.17	0.38 ± 0.15	0.779
	Size (nm^2^)	504.5 ± 3.4	531.7 ± 8.2	0.034	508.6 ± 3.3	492.7 ± 10.1	0.268
	Diameter (nm)	23.9 ± 0.4	25.5 ± 0.5	0.064	24.6 ± 0.8	24.3 ± 0.4	0.547
	TC (mg/dL)	117.6 ± 9.4	131.7 ± 6.6	0.159	133.7 ± 4.5	107.6 ± 4.1	0.591
	TG (mg/dL)	19.3 ± 1.3	17.6 ± 0.4	0.240	17.3 ± 0.8	17.7 ± 1.0	0.750

FI, fluorescence intensity (Ex = 370 nm, Em = 440 nm, 0.01 mg/mL of protein); MDA, malondialdehyde; TC, total cholesterol (μg/mg of protein); TG, triglyceride (μg/mg of protein); LDL, low-density lipoproteins; VLDL, very low-density lipoproteins. The data are expressed as the mean *±* SEM (standard error of the mean). The lipoprotein parameters in each group between weeks 0 and 12 were compared using a paired *t*-test.

## Data Availability

The data used to support the findings of this study are available from the corresponding author upon reasonable request.

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
