# Peer review of "Beneficial Effect of Cuban Policosanol on Blood Pressure and Serum Lipoproteins Accompanied with Lowered Glycated Hemoglobin and Enhanced High-Density Lipoprotein Functionalities in a Randomized, Placebo-Controlled, and Double-Blinded Trial with Healthy Japanese"

_ijms, 2023, doi:10.3390/ijms24065185_

Round 1

Reviewer 1 Report

The authors examined the impact of Cuban policosanol on blood pressure and various biochemical parameters with a focus on structure and function of lipoproteins in healthy Japanese. These and other authors addressed similar aims, however, in the non-japanese populations, which is highlighted as novelty of the present study.

Issues:

Paper is poorly written, there are too many grammar and syntax errors

which make reading and understanding difficult.

Page 12 bottom: In the description of the results shown in Fig.2 the authors wrote in the text LDL but the Fig shows VLDL. The whole description of the results presented in Fig 2 is very unclear and confusing; should be re-written.

Fig 2B legend: The text is not compatible with the graph.

Page 13; bottom, First sentence....(Fig. 1) - should be (Fig. 3).

Page 14; bottom: A comparison.... Fig 2A should be Fig. 4A...at week 12 (lane 1 ) should be lane 2....of week 0 (lane 2) should be lane 1.

Fig. 5 How many samples/gel lanes were used for densitometry? Why the authors did not make also native gels and TEM to be able to estimate HDL particle size; in the presented Fig 7, which should be Fig. 9, the authors indicated that Policosanol increases HDL particle size. However, particle size was not examined in the present study.

Fig. 6 Were PON activity measurements done in duplicates or triplicates for each serum sample; please indicate this in the legend. Please indicate as well whether the results are mean and SD or mean and SEM.

The figure showing zebrafish experiments is not Fig. 6 but Fig. 8.  For more clarity the legend to the Fig. 8A should be  slightly modified by adding HDL2 along with Policosanol or Placebo, for example:

CML + HDL2/Policosanol wk 0 (n=172)

CML + HDL2/Placebo wk 0 (n=170)  etc...

Page 18: In the presence of ..this sentence is completely gramatically wrong. The sentence says that HDL2 from the embryo shows the highest survivability. This is not correct; the embryos (not HDL )injected with HDL2 obtained from subjects treated with policosanol for 12  weeks showed the highest survivability. There many sentences througout the text in which  the meaning is altered due to poor grammar and syntax.

Abstract should be shorter.

Methods:

4.11 ....depending on the glycation of HDL2 and HDL3; why depending on glycation? please provide clarification.

Author Response

Thank you very much heartily for reviewing and critical comments to improve this paper.

Please find attached word file for our response, reflections, and revision in according to your valuable comments.

Reviewer 2 Report

The authors have investigated the putative beneficial effects of the consumption of Cuban policosanol on blood pressure, liver function, lipid parameters and HDL functionality in healthy Japanese volunteers.

The study design is poorly executed. Because of the low sample size, the statistical power is highly questionable. 30 and 35 participants were enrolled to follow up; however, 15 and 20 subjects were excluded because of the low compliance and only 15 / 15 subjects finalized the study successful. It is unclear, how were the low-compliance-subjects excluded. Significant improvement of blood pressure and liver functions found to be detected in the overall policosanol group (n=30) but this group also contained the low-compliance-subjects. However, changes of HDL-C and ApoAI levels were showed in the cleaned population.

The applied laboratory methods are very impressive. Although, several “gold standard” methods were performed i.e. sequential ultracentrifugation, lipoprotein gel electrophoresis, electron microscopic analysis, zebrafish embryo survivability; these methods are unnecessarily complicated and do not give adequate answer for the real question, how improves the policosanol the HDL functionality in vivo. The figure legends are incomplete and may mask the results.

The discussion does not support the results properly. The authors only mentioned own previous papers, the contradictory findings were not described (doi: 10.1093/ajcn/84.6.1543; doi: 10.1001/jama.295.19.2262). No limitations were included to the study.

Due to the above-mentioned reasons, I have doubts about the correctness of data; therefore, I do not suggest this manuscript for the publication.

Author Response

Thank you very much heartily for reviewing and critical comments to improve this paper.

Please find attached word file for our response, reflections, and revision in according to your valuable comments 

Round 2

Reviewer 1 Report

The authors improved the manuscript.

There are however still some issue that need to be addressed and improved:

1. The title is not correct. Improvement of blood pressure and hepatic function are not as the author say achieved via lowered glycated Hb and enhanced HDL function; the latter 2 are just accompanied by lower blood pressure and better liver function.

2. Abstract: second sentence: ...in improving lipid nad lipoprotein profiels; this is not the only scope of the study. In the first sentence the authors mentioned treatment of dyslipidemia and hypertension.Last sentence should be modified because blood pressure, heoatic functin and HbA1c are not improved VIA VLDL...I would suggest accompanied by improved structural and functional properties of serum lipoproteins.

The authors should more precisely say what was the major aim of the study, what they found and accordingly modify the title, abstract and conclusion.

I would suggest for title: Beneficial effect of Cuban policosanol on blood pressure and serum lipoproteins in healthy Japanese. I would not include liver function in the title and major aim because albumin, CHE and bilirubin are missing to have a complete picture on the liver. The abstract should be further shorthend; accori´ding to IJMS it should contain 200 words. The authors should extract most importamt results.

BUN is not an indicator of the liver function, rather renal function. Low BUN could even indicate decreased capacity of the liver to produce urea, which would mean impaired liver function. Increased ammonia would also indicate imapired liver function.  However, BUN rather reflects the extent of protein and water consuption as well as renal function.  Albumin serum levels or CHE activity in serum would be a good additional biomarkers of the biosynthetic activity of the liver. ALT, AST, y-GT  are markers of liver integrity/damage/ or overall health.

Limitation section in the Duscussion should be at the end before final conclusion.

Fig Legend to Fig. 8: please add Results are mean -/+ SD or SEM...

Author Response

Thank you very much heartily for reviewing and critical comments to improve this paper.

Please find attached doc as a response and reflections

Author Response

Thank you very much heartily for reviewing and critical comments to improve this paper

We accepted all your comments and suggestions

Please find attached doc as our response and reflections.

Round 3

Reviewer 2 Report

I wish you success in publishing your manuscript.